# B7-H3/CD276 Inhibitors: Is There Room for the Treatment of Metastatic Non-Small Cell Lung Cancer?

**DOI:** 10.3390/ijms232416077

**Published:** 2022-12-16

**Authors:** Umberto Malapelle, Paola Parente, Francesco Pepe, Martina Concetta Di Micco, Alessandro Russo, Celeste Clemente, Paolo Graziano, Antonio Rossi

**Affiliations:** 1Department of Public Health, University of Naples Federico II, Via S. Pansini, 80131 Naples, Italy; 2Pathology Unit, Fondazione IRCCS Ospedale Casa Sollievo della Sofferenza, Viale Cappuccini, 71013 San Giovanni Rotondo, Italy; 3Oncology Unit, Fondazione IRCCS Ospedale Casa Sollievo della Sofferenza, Viale Cappuccini, 71013 San Giovanni Rotondo, Italy; 4Medical Oncology Unit, A.O. Papardo, 98121 Messina, Italy; 5Oncology Centre of Excellence, Therapeutic Science & Strategy Unit, IQVIA, 20019 Milan, Italy

**Keywords:** B7-H3, CD276, DS-7300, enoblituzumab, immunotherapy, MGC018, NSCLC, obrindatamab

## Abstract

The striking clinical outcomes of antibody-based immunotherapy, through the inhibitors of cytotoxic T-lymphocyte-associated antigen-4 (CTLA-4) and the programmed cell death protein-1 (PD-1) and its ligand (PD-L1) axis, have driven research aimed at identifying further clinically relevant tumor antigens that can serve as targets in solid tumors. B7 homolog 3 protein (B7-H3, also known as CD276) is a member of the B7 family overexpressed in tumor tissues, including non-small cell lung cancer (NSCLC), while showing limited expression in normal tissues, becoming an attractive and promising target for cancer immunotherapy. B7-H3 expression in tumors has been demonstrated to be associated with poor prognosis. In addition to its role in immune modulation, B7-H3 also promotes pro-tumorigenic functions such as tumor migration, invasion, metastases, resistance, and metabolism. In this review, we will provide an overview of this newly characterized immune checkpoint molecule and its development in the management of metastatic NSCLC.

## 1. Introduction

The immune system works through the activation of several mechanisms, among which the immune checkpoints are the most investigated. The cytotoxic T-lymphocyte-associated antigen-4 (CTLA-4) and the programmed cell death protein-1 (PD-1) and its ligand (PD-L1) axis were the first immune checkpoints to be targeted with specific inhibitors (ICIs), which are now available in the clinical practice for the treatment of several tumors, including non-small cell lung cancer (NSCLC). These ICIs changed the natural history of non-oncogene-addicted NSCLC, becoming the standard of care for this subgroup of patients [1,2].

However, after a variable period of activity of anti-PD-1/PD-L1 and anti-CTLA-4 inhibitors, most patients show primary or acquired resistance to these ICIs. Thus, novel immune checkpoint targets, such as lymphocyte activation gene-3 (LAG-3), T cell immunoglobulin and mucin-domain containing-3 (TIM-3), T cell immunoglobulin and ITIM domain (TIGIT), V-domain Ig suppressor of T cell activation (VISTA), B7 homolog 3 protein (B7-H3), and B and T cell lymphocyte attenuator (BTLA), have emerged as promising therapeutic targets and are under clinical investigation [3].

B7-H3, also known as CD276, is a costimulatory/coinhibitory immunoregulatory protein that plays a dual role in the immune system during T cell activation and represents an attractive target for antibody-based immunotherapy [4].

In this review, we will provide an overview of this newly characterized immune checkpoint molecule and its development in the management of metastatic NSCLC.

## 2. Issues in Evaluation of B7-H3 Expression in Tissue Samples

B7-H3 expression in NSCLC has been widely investigated. The discrepancy concerning the relationship between B7-H3 expression and clinicopathological features or prognosis is well known and it could be attributable to several factors: the characteristics of the study population (smokers/nonsmokers; unresectable metastatic vs. locally advanced vs. resected NSCLC; prospective vs. retrospective analysis), the NSCLC histotype (adenocarcinoma vs. squamous cell carcinoma vs. small-cell lung cancer (SCLC)), the method to assess B7-H3 expression (flow cytometry (FC) vs. immunofluorescence (IF) vs. immunohistochemistry (IHC)), the B7-H3 clone used, the characteristics of the tissue analyzed (whole section vs. tissue macro arrays (TMA)), the B7-H3 expression in neoplastic vs. non-neoplastic cells/tumor immune microenvironment (TME), and the staining pattern (cytoplasmatic vs. membranous vs. both). Finally, there is no standardized method for the IHC scoring of B7-H3 expression (Table 1).

We choose to summarize the most recent studies on B7-H3 expression performed on human NSCLC in order to underline and analyze the major differences.

Yim et al. [5] investigated surface B7-H3 expression on tumor-infiltrating immune cells, neoplastic cells, and non-immune/non-neoplastic cells by FC in a prospective cohort of 84 patients, whereas the relationship between B7-H3 expression and clinicopathological features was explored in a retrospective cohort of 484 patients. On FC, surface B7-H3 expression was evaluated using the MIH42 clone in fresh tumor tissue matched to non-neoplastic lung tissue. In the formalin-fixed, paraffin-embedded (FFPE) whole section, IHC was performed using a D9M2L clone and B7-H3 immunostaining was quantified in tumor cells and non-tumor cells. On FFPE, B7-H3 expression was expressed as the percentage of tumor cells with complete/or incomplete membranous staining and designated as the tumor proportion score (TPS); non-tumoral B7-H3 expression status was classified as a non-tumoral proportion score (NTPS) according to the proportion of non-tumoral cells showing cytoplasmic and/or membranous staining. Interestingly, in all cellular subsets (TME, immune cells, and neoplastic cells), B7-H3 expression by FC was significantly higher in tumor tissue than in non-neoplastic tissue, suggesting its role in NSCLC development. High TPS (≥25%) on FFPE was associated with male gender and adverse clinicopathological features (squamous cell histotype, smoker status, wild-type epidermal growth factor receptor *(EGFR*) status, large neoplasm size, nodal metastasis, and poor tumor differentiation) and poor prognosis in patients with resected NSCLC. However, NSCLC with increased B7-H3 expression showed a higher number of tumor-infiltrating CD45+ immune cells, natural killer (NK) cells, and CD8+ T cells, suggesting a prognostic role for B7-H3 in immunotherapy [5].

Mao et al. [6] investigated B7-H3 expression by IHC in 128 resected NSCLC patients to evaluate the relationship between B7-H3 expression, clinicopathologic variables, and prognosis. Immunostaining was performed using a mouse anti-human B7-H3 mAb (Clone 4H7). B7-H3 expression was indicated as the percentage of tumor cells showing cytoplasmatic or membranous immunoreactivity as follows: 0 = no staining, 1 = 1–10% of cells, 2 = 11–50% of cells, and 3 = more than 50% of cells stained. Staining was scored using a semiquantitative method as follows: 0 = no staining, 1 = weak staining, 2 = moderate staining, and 3 = strong staining. The immunoreactive score (IRS) was calculated by multiplying the tumor cell staining intensity and the percentage of positively stained neoplastic cells. IRS ranged from 0 to 9, whereby 0 indicated no staining (e.g., negative results) and 1, 2, or 3 and more than 3 corresponded to weak, moderate, and strong expression, respectively. In the final analysis, no or weak staining was considered as negative expression, whereas moderate or strong staining was considered as positive expression. This study demonstrated a direct correlation between high B7-H3 expression and lymph node metastasis (*p* < 0.05) and TNM staging [6].

A similar scoring system with a different H7-B3 clone (SP206) was applied by Yang et al. [7] in 56 advanced pulmonary adenocarcinoma patients harboring activating *EGFR* mutations. Briefly, the intensity of membranous and cytoplasmatic B7-H3 reactivity in neoplastic cells was defined as 0 (no staining), 1 (weak), 2 (intermediate), and 3 (strong). The percentage of positive tumor cells was categorized as follows: 0 (<5%), 1 (5–25%), 2 (26–50%), 3 (51–75%), and 4 (>75%). An overall immunoreactive score was assigned by multiplying the staining intensity and the percentage of positive tumor cells, which yielded a range from 0 to 12. All cases were divided into B7-H3 low (<6) and B7-H3 high groups (6–12). The B7-H3 high group was associated with poor progression-free survival (PFS) and overall survival (OS), suggesting a predictive role of B7-H3 in patients with activating EGFR mutations. However, the distribution of high and low B7-H3 expression was not related to age, sex, tumor size, staging, or *EGFR* mutation patterns [7].

In order to evaluate the prognostic association between B7-H3 expression levels, smoking history, and clinicopathological/molecular features, Inamura et al. [8], using clone BD/5A11, described immunohistochemical B7-H3 expression in TMA representative of 270 consecutive cases of lung adenocarcinoma. Only membranous staining in tumoral cells was assessed. The intensity of B7-H3 membranous expression (B7-H3 intensity) in cancer cells was defined as 0 (absent), 1 (weak to moderate), or 2 (strong). The percentage of tumor cells with each B7-H3 intensity was scored and subclassified into two groups as follows: low B7-H3 expression group (intensity 1 < 50% and intensity 2 < 10%) and high B7-H3 expression group (intensity 1 ≥ 50% or intensity 2 ≥ 10%). High B7-H3 expression was correlated with adverse prognosis and decreased OS in moderate/heavy-smoking patients. Moreover, high B7-H3 expression was associated with tumor grading, wild-type *EGFR* status, and staging [8].

The same B7-H3 clone and a similar IHC scoring system were applied by Yonesaka et al. [9] in 82 advanced or recurrent NSCLC patients, of which 50 patients were treated with anti-PD-1 therapy. The B7-H3 staining pattern was scored as 0, 1+, 2+, or 3+ as follows: 0, no membranous staining, ≤10% tumor cells with faint/weak membranous staining; 1+, >10% tumor cells with faint/weak membranous staining; 2+, >10% tumor cells with weak or moderate membranous staining or ≤10% tumor cells with strong membranous staining; 3+, >10% tumor cells with strong membranous staining. Interestingly, the B7-H3 0 staining pattern showed the highest response rate to PD-1 therapy at 88%, whereas B7-H3 expression was associated with refractoriness [9].

Boland et al. [10] described B7-H3 immunostaining performed on a large series of 214 squamous lung cancer samples. Immunostaining performed by the AF1027 clone was demonstrated in the whole section only in tumoral cells and defined as negative (0% of tumor cells staining) or positive. In positive cases, an estimate of the overall percentage of immunopositive cells was determined as follows: 1% positivity for only rare cell staining; otherwise, the percentage of immunopositive cells was estimated by using 5% increments, 5–100%. Only membranous and circumferential staining was considered positive. The intensity of staining was also estimated as weak (faint brown staining), moderate, or strong (dark brown staining). In detail, 189 out of 214 samples (88.3%) showed a positive membrane signal for B7-H3 expression. The co-expression of B7-H3 and B7-H1/PD-L1 was documented in 40 cases, with a weak–moderate B7-H3 intensity. Remarkably, no staining signal was detected for B7-H1/PD-L1 or B7-H3 in control arm samples or in surrounding lymphocytes. In addition, the IHC signal was remarkably related to the prognosis of tumor patients [10].

Altan et al. [11] developed a computerized scoring system based on the pixel intensity (AQUA^®^ method) of quantitative immunofluorescence (QIF) in the neoplastic and stromal compartment. B7-H3 expression was induced with the D9M2L clone in both membranous and cytoplasmatic staining on 634 TMA from NSCLC patients. The authors documented the high expression of B7-H3 in the majority of NSCLC cases and it was associated with smoking history. Overexpression of B7-H3 protein (highest 10%) was correlated with poor survival. Interestingly, an exclusive pattern with infrequent co-expression and co-localization between B7-H3 and B7-H1/PD-L1 protein was observed. However, this method was poorly reproducible in pathology laboratories [11].

## 3. The Target

CD276 consists of a type I transmembrane protein composed of two immunoglobulin constant (IgC) and variable (IgV) domains in extracellular domains (Figure 1) [12].

This protein shows controversial effects on the activation of CD4+ and CD8+ immune cells. In particular, authors observed an improvement in cytotoxic T cells’ activation and interferon gamma (IFNγ) production under CD276 signal regulation [13], while a tumor escaping from immune cell surveillance was also identified by activating NFAT, NF-kB, and AP-1-related molecular pathways [8,14]. Given these results, the uncertain role played by CD276 was considered dependent on co-regulation with a plethora of different molecules, including PD-1, CTLA-4, or different types of immune response-related cells [15]. Notably, it was demonstrated that CD276 mRNA levels were overexpressed in various types of cancer in comparison with matched normal tissues [8,16]. In another study, Yu et al. focused on the prognostic evaluation of CD276 expression levels in metastatic NSCLC [17]. In particular, they observed a significant increase in terms of proliferation, invasion, and migration rate in in vitro systems. In addition, a statistically relevant association was observed between the B7-H3 high expression group and TNM stage. Specifically, the metastasis rate (lymph nodes and distant metastatic sites) was significantly associated with the B7-H3/CD276 expression level (*p* > 0.05) [17]. Similarly, Yim et al. highlighted a significantly high B7-H3 expression level in NSCLC patients that harbored low prognosis markers (nodal metastasis, poor differentiation). In a retrospective series of *n* = 484 NSCLC patients, the authors observed poor 5-year overall survival in the highly expressed CD276 group (as previously described) [5]. Accordingly, the heterogeneous expression in human tissue of this promising biomarker suggests that post-transcriptional modifications regulate protein expression levels [18]. Considering that lung cancer is one of the most prevalent causes of death worldwide [19], ICIs have showed a promising effect in the treatment of a non-negligible percentage of advanced lung cancer patients [20]. However, resistance mechanisms reduce the clinical benefit of this approach. Moreover, several studies have investigated the role of CD276 as a potential target to overcome ICI resistance. Yang et al. evaluated how targeting this biomarker may overcome resistance to ICIs. In fact, they engineered natural killer receptors with a chimeric antigen in order to enhance the drug response in cell systems [21].

The molecular landscape of lung cancer patients appears complex and heterogeneous. The identification of the connection between B7-H3 expression and other molecular actors is currently under investigation. In this context, the implementation of Next-Generation Sequencing (NGS technologies) sheds light on these aspects. In particular, Nakagomi et al. evaluated genomic hallmarks that could explain sensitive/resistance mechanisms for therapeutic agents against B7-H3 [22]. Remarkably, the authors observed a different molecular profile between mucinous and non-mucinous lung adenocarcinoma patients that harbored distinct molecular patterns according to *KRAS*, *EGFR*, and *p53* molecular alterations detected by using an NGS platform. Moreover, IHC also highlighted B7-H3 differentially expressed in 42.4% and 19.4% of invasive mucinous and non-mucinous ADC patients. These data elucidated the necessity of integrating technical approaches able to correlate molecular findings with pathological data in order to predict clinical outcomes in lung cancer patients. In a non-negligible percentage of cases, diagnostic specimens represent a “scant” source of material for histological diagnosis and molecular analysis in advanced lung cancer patients [23]. In this scenario, liquid biopsy specimens, which consist of peripheral blood withdrawal, where several analytes may be purified and investigated, may represent an integrative diagnostic tool when tissue specimens are not available in advanced tumor stages. In particular, blood specimens may also be considered a reliable option to detect ICI biomarkers. Several groups showed promising results for the analysis of PD-L1 expression levels by comparing blood-based results with matched tissue specimens. Weber et al. demonstrated a remarkable PD-L1 expression level in metastatic lung cancer patients detected by a RTqPCR approach. Briefly, the authors detected a statistically relevant variation in the PD-L1 mRNA expression level between healthy volunteers and liquid biopsy-based molecular analysis in tumor patients. In addition, cfDNA revealed a prognostic role according to the highest mRNA expression level in metastatic (N+) lung cancer patients. In addition, they also evaluated the non-inferiority of blood-based analysis with respect to the molecular analysis of conventional tissue specimens [24]. Similarly, Zhang et al. focused on the soluble B7-H3 form (sB7-H3) in order to evaluate the clinical significance of this approach for clinical purposes. The authors adopted an ELISA assay to detect sB7-H3 in peripheral blood specimens from NSCLC patients. Interestingly, the authors identified a positive result in terms of sB7-H3 levels in tumor patients with respect to healthy donors (*p* < 0.001) or non-malignant pulmonary disease patients (*p* < 0.001). Of note, the authors set a clinical cutoff of 30 ng/µL, able to distinguish tumors from non-malignant patients and healthy individuals (sensitivity of 48.8 and 48.0%; specificity 98.5 and 93.7%, respectively). In addition, increasing sB7-H3 levels correlated with clinical data better than conventional prognostic biomarkers, including CEA, CA19-9, and CA153 [25].

Remarkably, the B7-H3 expression level is emerging as a promising biomarker to identify novel resistance mechanisms for ICI administration. In particular, both the mRNA expression levels and protein abundance may be considered targets for novel drugs that could overcome ICI resistance. In this scenario, novel technical approaches able to identify expression signatures in diagnostic routine samples play a pivotal role in the clinical stratification of tumor patients from a widespread series of biological matrices [11,22,24].

## 4. The Inhibitors

Recent knowledge in molecular biology and advances in antibody engineering have enabled the targeting of B7-H3 through several mechanisms. Among these, antibody–drug conjugates, mAbs mediating cellular cytotoxicity, and CD3-engaging bispecific antibodies are the therapeutic approaches being investigated in phase I/II trials in solid tumors, with preliminary results already available also for the treatment of NSCLC (Table 2 and Table 3; Figure 1 and Figure 2) [26].

### 4.1. Antibody–Drug Conjugates

The antibody–drug conjugates (ADCs) combine the target specificity of a monoclonal antibody (mAb) with cytotoxic agents and are delivered to a tumor, improving the therapeutic index (Figure 2).

Among the main ADCs targeting B7H3, MGC018 is a humanized B7-H3 mAb with a cleavable linker–duocarmycin payload, which delivers duocarmycin to tumors. A phase I/II, first-in-human trial is assessing its safety when administered alone or in combination with an anti-PD-1 mAb in solid tumors, including NSCLC, expressing B7-H3 (NCT03729596).

DS-7300a is another B7-H3-specific mAb conjugated to an exatecan derivative payload with around four topoisomerase I inhibitor particles. A dose escalation (part 1) and expansion (part 2) study is recruiting patients with advanced solid tumors, including NSCLC. DS-7300 is administered at doses ranging from 0.8 to 16 mg/kg, intravenously, every 3 weeks. Part 1 is assessing the safety, tolerability, and maximum tolerated or recommended dose (MTD) for the expansion part of the study, in which additional patients are being enrolled to examine the safety and efficacy. Preliminary results from this phase 1/2 trial showed, among the 147 previously pretreated patients with solid tumors enrolled, a baseline median B7-H3 membrane H-score by IHC of 185 (0–300), with no dose-limiting toxicity (DLT). Treatment-emergent adverse events (TEAEs) occurred in 144 patients (98%). The most common all-grade TEAEs were nausea (63%), infusion-related reactions (IRRs; 32%), anemia (33%), decreased appetite (31%), and vomiting (30%). Grade ≥ 3 TEAEs occurred in 66 patients (45%), with one (1%) TEAE leading to death, and 11 patients (8%) had a dose discontinuation. A total of 33 confirmed partial responses (28%) were reported out of 118 evaluable patients. Nine squamous NSCLC patients were enrolled, reporting two (40%) confirmed partial responses out of the five evaluable patients. Overall, DS-7300 was generally well tolerated, with interesting clinical activity in heavily pretreated patients with advanced solid tumors [27].

### 4.2. mAbs Mediating Cellular Cytotoxicity

Antibody-dependent cell-mediated cytotoxicity (ADCC) refers to antibodies that can bind to their specific antigens on the malignant cell, via their antigen-binding fragment (Fab) portions, and interact with effector cells via their fragment-crystallizable region (Fc) portions, thereby acting as bridges that link the effector to a target. Moreover, in order for an effector cell to carry out ADCC, Fc can bind to Fc receptors (FcR) on the surfaces of killer cells to mediate the direct killing of target cells (Figure 2) [29].

Enoblituzumab (MGA271) is a fully humanized mAb, bearing an Fc domain engineered to enhance its antitumor function by increasing binding to the activating receptor, CD16A, and reducing it to the inhibitory receptor, CD32B. Enoblituzumab was the first mAb tested against B7-H3-expressing tumors, being effective in multiple cancer types through antibody-dependent cellular cytotoxicity [30].

A phase I clinical trial enrolled 179 pretreated cancer patients, including melanoma, prostate cancer, bladder cancer, breast cancer, clear cell renal carcinoma, NSCLC, and head and neck squamous cancer cell (HNSCC), showing that enoblituzumab, as a single agent, up to the dose of 15 mg/kg weekly, was well tolerated, with no DLT or MTD. Disease stabilization (>12 weeks) and tumor shrinkage (2–69%) were seen across the several tumor types enrolled [31].

A phase I/II study considered patients affected by urothelial cancer, cutaneous melanoma, HNSCC with or without prior PD-(L)1 treatment, and NSCLC with or without previous PD-(L)1 therapy, who received intravenous enoblituzumab (3–15 mg/kg) weekly plus intravenous pembrolizumab (2 mg/kg) every 3 weeks during dose escalation and cohort expansion. A total of 133 patients were enrolled, with no MTD reached. Grade ≥ 3 treatment-related adverse events occurred in 28.6% of patients, with one treatment-related death due to pneumonitis. The outcomes were interesting in patients who were ICI-naïve. In fact, the objective response rate (ORR) was 33.3% in the 18 patients with HNSCC who were naïve for ICI therapy, and 35.7% in the 14 patients with ICI-naïve NSCLC, with median progression-free survival (PFS) of 3.48 and 4.83 months and median overall survival (OS) of 17.38 and 12.32, respectively [28].

The phase II CP-MGA271-06 trial investigating enoblituzumab plus retifanlimab or tebotelimab as a first-line treatment for patients with recurrent or metastatic HNSCC closed early due to safety concerns, with seven fatalities potentially linked to hemorrhagic events [32].

### 4.3. CD3-Engaging Bispecific Antibodies

Bispecific antibodies (BsAbs) are composed of the fragments of two distinct Abs, wherein one arm can bind to the CD3 component on T cells whereas the other arm recognizes a tumor-specific antigen, such as B7-H3 on tumor cells. In this way, T cells are recruited to the tumor site and activated to kill cancer cells [33].

Obrindatamab (MGD009) is a humanized CD3xB7-H3 dual-affinity protein investigated in previously treated solid B7-H3-expressing tumors. After the Food and Drug Administration (FDA) suspended the study for safety reasons, due to hepatic adverse events, and it subsequently reactivated, because these events were uncomplicated and short-lived, the trial was permanently terminated due to a business decision (NCT02628535).

## 5. Conclusions

*What is known?* B7-H3 is a novel immune checkpoint from the B7 family that is highly expressed in NSCLC. Interestingly, B7-H3 expression in NSCLC seems to be correlated with low PD-L1 expression and with a poor prognosis in *EGFR* mutant patients, leading to speculation about an alternative biological pathway and suggesting a role for B7-H3 as an ICI in NSCLC patients [7,9,10]. Compared to other ICIs, B7-H3 appears to be a unique and powerful target, having a role in immune-mediated and non-immune-mediated pathways.

*What is not known?* We are still in the early stage of understanding this new potential immune system target, with the aim to include B7-H3 in the clinical practice of NSCLC care. The change practice successes of anti-PD-(L)1 and anti-CTLA4 agents in the management of advanced non-oncogene-addicted NSCLC [1,2] are examples that offer a promising direction to develop new target immunotherapies for B7-H3. Ongoing phase II/II studies will clarify any side effects of B7-H3 use.

*What has already been learned from the use of B7-H3 inhibitors*? Preliminary results from several different B7-H3-based cancer immunotherapy strategies in NSCLC come mainly from antibody–drug conjugates, mAbs mediating cellular cytotoxicity, and CD3-engaging bispecific antibodies strategic approaches.

Results from a phase I/II study [28] are encouraging in patients who are ICI-naïve, with an objective response rate (ORR) of 35.7% in NSCLC and PFS of 4.83 months and OS of 12.32, respectively.

*What is needed to apply and explore it in the future*? Overall, a deeper biological tumor profile in NSCLC, including B7-H3 assessment, is imperative. In this setting, shared guidelines concerning B7-H3 clones and scoring systems (CPS vs. TPS vs. other) in NSCLC tissue are needed in order to better schedule appropriate ICI therapy in NSCLC. Further, clear data on B7-H3 expression in metastases are needed to design B7-H3-targeting therapies. Among the additional in-depth investigations, the assessment of the potential diagnostic and prognostic value of the B7-H3 serum level as a monitoring marker should be also considered.

Other therapeutic strategies are being evaluated, such as bi- and tri-specific killer engagers, the chimeric antigen receptor (CAR) T cell technology, and radio-immunotherapy, which are in preclinical, in vitro and in vivo, and very early clinical stages of investigation in solid tumors [34]. mRNA levels and protein expression are currently under investigation as integrated biomarkers for the identification of novel therapeutic strategies in ICI-resistant NSCLC patients [24,35]. This approach underlines how different biological matrices may be integrated with molecular data in the concept of “3D biology”, where DNA, RNA, and protein-related information can be systematized to obtain a comprehensive molecular profile for NSCLC patients [36].

## Figures and Tables

**Figure 1 ijms-23-16077-f001:**
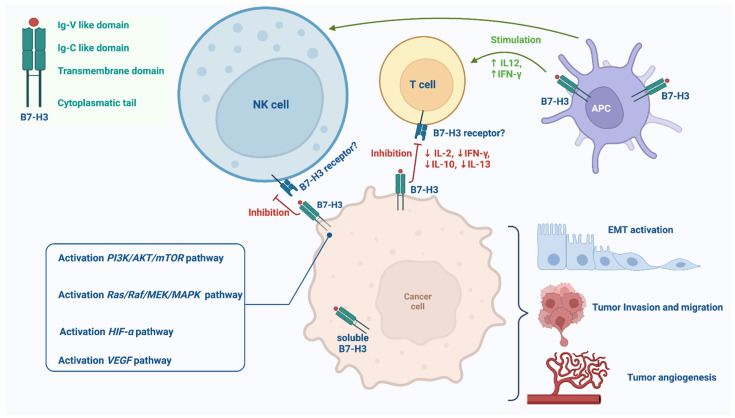
B7-H3 non-immune-mediated (box in the lower left corner) and immune-mediated (cartoon) signaling pathways. *PI3K/AKT/mTOR* and *Ras/Raf/MEK/MAPK* signaling pathways are involved in promoting the migration, invasion, and epithelial–mesenchymal transition (ETM) of cancer cells. *HIF-α* pathway is involved in glucose metabolic reprogramming and enhancing neoplastic tumor growth. *VEGF* signaling pathway is involved in promoting neo-angiogenesis and metastasis. B7-H3 acts as immune co-stimulatory molecule, increasing IFN-γ and IL-12 levels and promoting CD4+ and CD8+ T cells’ proliferation and enhancing cytotoxic T cell activity. Moreover, B7-H3 plays an immune co-inhibitory role, reducing cytokines (IL-2, IL-10, IL-13, and IFN-γ) and inhibiting T cell proliferation and NK cell activity.

**Figure 2 ijms-23-16077-f002:**
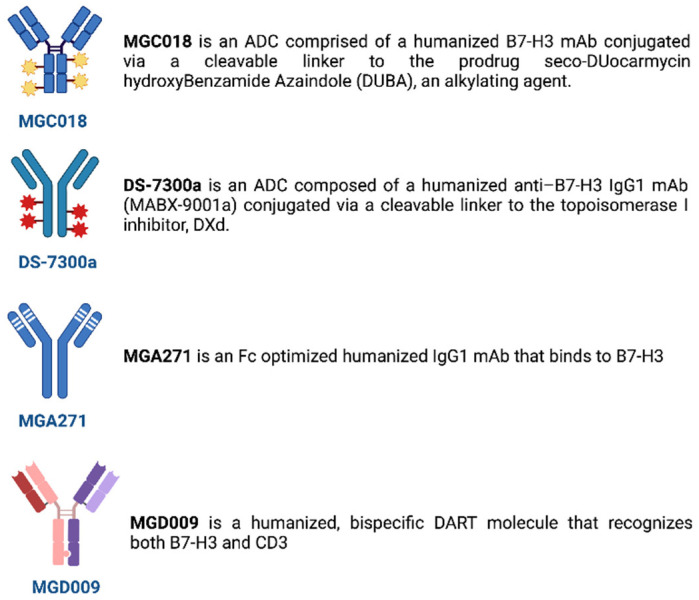
Main B7-H3 inhibitors in solid tumors. ADC: antibody–drug conjugates; mAb: monoclonal antibody; Fc: crystallizable region.

**Table 1 ijms-23-16077-t001:** Study on human NSCLC with clinicopathological findings and analytical methods applied.

B	Study	Patients	Histotype	Oncology	Specimen	Method	Clone	Cellularity	SP	Scoring System
5	P	84	AC+SCC	Re	WS	FC	MIH42	N+NN	M	% positive cells
	R	484	AC+SCC	Re	WS	IHC	D9M2L	N	M	TPS
								NN	M+C	NTPS
6	R	128	AC+SCC	Re	WS	IHC	4H7	N	M+C	IRS
7	R	56	AC	UR; m*EGFR*	WS	IHC	SP206	N	M+C	IRS
8	R	270	AC	Re	TMA	IHC	BD/5A11	N	M	Staining intensity + percentage of positive cells
9	R	82	AC+SCC	UR/Me/RE	WS	IHC	BD/5A11	N	M	Staining intensity + percentage of positive cells
10	R	214	SCC	Re	WS	IHC	AF1027	N	M	Staining intensity + percentage of positive cells
11	R	634	AC+SCC	Re	TMA	QIF	D9M2L	N+NN	M+C	QIF

B: bibliography; P: prospective; R: retrospective; AC: adenocarcinoma; SCC: squamous cell carcinoma; Re: resectable; UR: unresectable; Me: metastatic; RE: recurrent; m*EGFR*: activating epidermal growth factor receptor mutations; TMA: tissue macro array; WS: whole section; IF: immunofluorescence; IHC: immunohistochemistry; QIF: quantitative immunofluorescence; N: neoplastic cells; NN: non-neoplastic cells; SP: staining pattern; M: membranous; C: cytoplasmatic; TPS: tumor proportional score; NTPS: non-tumor proportional score; IRS: immunoreactive score—percentage of immunostained cells x staining intensity.

**Table 2 ijms-23-16077-t002:** Main ongoing clinical studies investigating B7-H3 inhibitors in solid tumors including NSCLC [26].

Protocol ID	Title	NSCLC Eligibility	Primary Endpoint	Status
NCT03729596	A phase 1/2, first-in-human, open-label, dose-escalation study of MGC018 (anti-B7-H3 antibody drug conjugate) alone and in combination with MGA012 (anti-PD-1 antibody) in patients with advanced solid tumors	Yes	AEs, MTD	Recruiting
NCT05293496	A phase 1/1b dose escalation and cohort expansion study of MGC018 in combination with checkpoint inhibitor in participants with advanced solid tumors	No	AEs, SAEs	Recruiting
NCT04145622	Phase 1/2, two-part, multicenter first-in-human study of DS-7300a in subjects with advanced solid malignant tumors	Yes	DLT, AEs	Recruiting [27]
NCT02381314	A phase 1, open-label, dose escalation study of MGA271 in combination with ipilimumab in patients with melanoma, non-small cell lung cancer, and other cancers	Yes	AEs	Completed
NCT03406949	A phase 1, open label, dose escalation study of MGD009, a humanized B7-H3 x CD3 DART^®^ protein, in combination with MGA012, an anti-PD-1 antibody, in patients with relapsed or refractory B7-H3-expressing tumors	Yes	TEAEs, MTD/MAD	Completed

NSCLC: non-small cell lung cancer; AEs: adverse events; MTD: maximum tolerated dose; SAEs: serious adverse events; DLT: dose limiting toxicity; TEAEs: treatment-emergent adverse events; MAD: maximum administered dose.

**Table 3 ijms-23-16077-t003:** Available clinical results on B7-H3 inhibitors in NSCLC.

Study	Phase of Study	Treatment	No. Pts	ORR (%)	PFS (Months)	OS (Months)	Safety (All Population)
Doi T, et al. [27]	I/II	DS-7300	70	21.4	NR	NR	TEAEs: 98.6% Grade ≥ 3 TEAEs: 31.4% Serious TEAEs: 21.4% TEAEs leading to death: 2.9%
			NSCLC: 4	25			
Aggarwal C, et al. [28]	I/II	Enoblituzumab + Pembrolizumab	133	14.7 (102 evaluable pts)	NR	NR	TRAEs: 87.2% Grade ≥3 TRAEs: 28.6% TRAEs leading to death: 0.75%
			NSCLC ICI-naïve: 14	35.7	4.83	12.32	
			NSCLC prior ICIs: 21	9.5	3.45	7.13	

NSCLC: non-small cell lung cancer; No. Pts: number of patients; ORR: objective response rate; PFS: progression-free survival; OS: overall survival; NR: not reported; ICIs: immune checkpoint inhibitors; TEAEs: treatment-emergent adverse events; TRAEs: treatment-related adverse events.

## Data Availability

Not applicable.

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
