# Peer review of "B7-H3/CD276 Inhibitors: Is There Room for the Treatment of Metastatic Non-Small Cell Lung Cancer?"

_ijms, 2022, doi:10.3390/ijms232416077_

Round 1

Reviewer 1 Report

After evaluating the manuscript “B7-H3/CD276 inhibitors: is there a room for the treatment of metastatic non-small cell lung cancer?" I have to recommend its minor revision in current version.

Minor

1. For clarity, please add a figure with the chemical formulas of the inhibitors discussed in section 4 (in particular, in table 2).

Author Response

Reviewer’s answers:

After evaluating the manuscript “B7-H3/CD276 inhibitors: is there a room for the treatment of metastatic non-small cell lung cancer?" I have to recommend its minor revision in current version.

Minor

For clarity, please add a figure with the chemical formulas of the inhibitors discussed in section 4 (in particular, in table 2).

We are very grateful to Reviewer for her/his kind considerations about our manuscript and for her/his useful suggestions that allow us to improve our paper.

The text has been revised according to Reviewer’s comments and we added Figure 2 based on Reviewer’s suggestion.

We hope the required integrations may be deemed satisfactory and exhaustive.

Reviewer 2 Report

Malapelle et al highlighted B7-H3 expression in tumors is associated with poor prognosis in NSCLC, and the potential effect of B7-H3 inhibitors to treat metastasis NSCLC. Overall, the review is well performed, but there are some major points the author would need to address.

Major points:

1. The review title is “B7-H3/CD276 inhibitors: is there a room for the treatment of 2 metastatic non-small cell lung cancer?” However, overall didn’t highlight the connection of B7-H3 with lung cancer metastasis, or lung cancer cell migration, and infiltration. Need more correlated papers or discussion on this point.

2. Instead of giving a lot of details of each study, such as line 91-94, line 121-124, line 131-134, line 260-267, line 292-298……, the author should give the clear general description of the results, not just use the results from the original paper. The author need to give the comprehensive conclusion from the published papers, and tell the reader what is known, what is the not known, what is already learned from the lesson of B7-H3 inhibitors, what is need to explore and apply in the future?

3. The figure legend of figure 1 need to describe more details. What are connections between NK or T cells and tumor cells through B7-H3?

Author Response

Reviewer’s answers:

Malapelle et al highlighted B7-H3 expression in tumors is associated with poor prognosis in NSCLC, and the potential effect of B7-H3 inhibitors to treat metastasis NSCLC. Overall, the review is well performed, but there are some major points the author would need to address’.

We are very grateful to Reviewer for her/his kind considerations about our manuscript and for her/his useful suggestions that allow us to improve our paper.

The text has been revised according to Reviewer’s comments as follows:

Major points:

  1. The review title is “B7-H3/CD276 inhibitors: is there a room for the treatment of metastatic non-small cell lung cancer?” However, overall didn’t highlight the connection of B7-H3 with lung cancer metastasis, or lung cancer cell migration, and infiltration. Need more correlated papers or discussion on this poin’t.

We agree with the Reviewer’s comment. We added a paragraph with the appropriate requests (lines 192-201 of the revised version) and a more detailed description in the legend of the figure 1 with relative references.

  1. Instead of giving a lot of details of each study, such as line 91-94, line 121-124, line 131-134, line 260-267, line 292-298……, the author should give the clear general description of the results, not just use the results from the original paper.

We thank the Reviewer for her/his suggestions. We tried to improve all sentences underlined (lines 68-88, 113-128, 232-236, 248-254 of the revised version)

  1. The author need to give the comprehensive conclusion from the published papers, and tell the reader what is known, what is the not known, what is already learned from the lesson of B7-H3 inhibitors, what is need to explore and apply in the future?

We agree with the Reviewer’s very fine suggestion. We revised paragraph 5 in Conclusion, with point-by-point state of art about B7-H3.

  1. The figure legend of figure 1 need to describe more details. What are connections between NK or T cells and tumor cells through B7-H3?

We agree with the Reviewer for her/his appropriate suggestion. We modified the figure legend of the figure 1 in a more detailed way.

We hope the required integrations may be deemed satisfactory and exhaustive.

Round 2

Reviewer 2 Report

No more comments.